

# Comparative Assessment of SSR and RAPD markers for genetic diversity in some Mango cultivars

Mohammed A. A. Hussein[1], Manal Eid[1], Mehdi Rahimi[2], Faten Zubair Filimban[3] and Diaa Abd El-Moneim[4]

[1] Department of Botany (Genetic Branch), Faculty of Agriculture, Suez Canal University, Ismailia, Egypt

[2] Department of Biotechnology, Institute of Science and High Technology and Environmental Sciences, Graduate University of Advanced Technology, Kerman, Iran

[3] Division of Botany, Department of Biology, Faculty of Sciences, King Abdulaziz University, Jeddah, Saudi Arabia

[4] Faculty of Environmental Agricultural Sciences, Plant Production Department - Genetic Branch, Arish University, El Arish, Egypt

## ABSTRACT

Genetic improvement mainly depends on the level of genetic variability present in the population, and the degree of genetic diversity in a population largely determines the rate of genetic advancement. For analyzing genetic diversity and determining cultivar identities, a molecular marker is a useful tool. Using 30 SSR (simple sequence repeat) and 30 RAPD (randomly amplified polymorphic DNA) markers, this study evaluated the genetic divergence of 17 mango cultivars. The effectiveness of the two marker systems was evaluated using their genetic diversity characteristics. Additionally, the effects of SM (simple matching) and Dice similarity coefficients and their effects on mango clustering were evaluated. The findings showed that SSR markers generated 192 alleles, all of which were polymorphic (100%). With RAPD markers, 434 bands were obtained, 361 of which were polymorphic (83%). The average polymorphic information content (PIC) for RAPD and SSR was 0.378 and 0.735, respectively. Using SSR markers resulted in much higher values for other genetic diversity parameters compared to RAPD markers. Furthermore, grouping the genotypes according to the two similarity coefficients without detailed consideration of these coefficients could not influence the study results. The RAPD markers OPA_01, OPM_12 followed by OPO_12 and SSR markers MIAC_4, MIAC_5 followed by mMiCIR_21 were the most informative in terms of describing genetic variability among the cultivars under study; they can be used in further investigations such as genetic mapping or marker-assisted selection. Overall, 'Zebda' cultivar was the most diverse of the studied cultivars.

# INTRODUCTION

The mango fruit (*Mangifera indica* L.) is one of the most nutritious and expensive edible fruits globally. It has a diploid genome (2n = 2x = 40 chromosomes) and belongs to the order Sapindales, family Anacardiaceae, and genus *Mangifera*. Mangoes are grown on

Corresponding author
Diaa Abd El-Moneim,
dabdelmoniem@aru.edu.eg

2.5 million ha in tropical and subtropical regions of the world, with an annual production of approximately 46.6 million tons (*Wang, Luo & Huang, 2020*). The mango is the fifth most essential fruit crop, followed by bananas, grapes, apples, and oranges (*Deshpande et al., 2017*).

The mango is the most exquisite fruit in Egypt and one of the world's best fruits. Mangoes have been planted in 289,020 feddans (1 feddans = 0.42 hectare) in Egypt, according to statistics from the Ministry of Agriculture and Land Reclamation (2020), with a maximum annual yield of two million and 800 tons.

Several varieties with different origins grow in Egypt. From India and Sri Lanka come the 'Hindi Besennara', long, 'Banarasi Langra', and 'Mabrouka' varieties, and from Florida and South Africa come the 'Carrie', 'Glenn', 'Keitt', and 'Kent' varieties. The local varieties include the 'Zebda', 'Taymour', 'Mesk', 'Senarry', 'Mstikawy', and 'Dabsha' mangoes (*Abdelsalam et al., 2018*).

The traditional methods of identifying or distinguishing cultivars have used the morphological traits of the plant's leaves, flowers, and fruit (*Avramidou et al., 2023*). This approach has been unsuccessful since closely related cultivars often cannot be differentiated by these traits and since environmental factors can affect the expression of the traits (*e.g.*, climate conditions or cultivation procedures may have impacts).

Genetic variation in the germplasm of plants is a crucial factor in new cultivar creation because inbreeding in cultivated plants leads to a quick loss in vigor, yield, and fruit size (*Spangelo et al., 1971*). For these reasons, modern breeders need more efficient methods of breeding that are rapid, informative, and unaffected by environmental factors. Molecular markers have been replacing or complementing traditional morphological and agronomic characterizations since they are extremely plentiful, cover the genome, and are not influenced by the environment. In the case of fruit trees with a long juvenile period, these markers can make the characterization of new cultivars less time-consuming (*Ramadasappa, Mallaiah & Ramakrishnaiah, 2021*).

Previous studies have found that molecular markers are unlimited in number, remain unaffected by the environment and growth conditions, and are simply inherited (*Karihaloo et al., 2003*). Various molecular markers such as the randomly amplified polymorphic DNA (RAPDs) of *Adhikari & Sinchan (2015)*, amplified fragment length polymorphisms (AFLPs) of *Eiadthong et al. (2000)*, and simple sequence repeats (SSRs) of *Bukhari et al. (2022)* have been tested for genetic diversity assessment in mangoes. There has been increasing development and generalized use of many methodologies during recent years, and now comparative studies are needed to choose the best DNA marker technology for fingerprinting and diversity studies in terms of reproducibility, cost, sensibility, and level of polymorphism detection. One technique may be more appropriate than another in a study, and different techniques may be informative at different taxonomic levels. Similarities between the different molecular techniques have begun to be debated, but the results have conflicted among authors. Several works have reported comparable results among different markers (*Bally et al., 2021*; *Luo et al., 2011*; *Powell et al., 1996*; *Srivastava et al., 2012*).

Although molecular markers only reflect a small portion of a plant's genome, they are used to infer links between the complete genome among a number of populations. The precision of resulting estimations of genetic distance will depend on how the loci discovered by individual marker analysis approaches are distributed (*Nei, 1987*). Therefore, it is ideal for assessing genetic diversity because the loci discovered are scattered randomly throughout and must be a sample to reflect the entire genome. Genetic maps are required to compare the distribution of various individual markers, but this takes much time. As a result, an alternate strategy has been put out to calculate the relationships between the germplasm accessions produced from various marker approaches utilizing statistics analysis and various similarity coefficients (*Liu, 1998*). The molecular marker data have been compared using different similarity coefficients and clustering methods (*Meyer et al., 2004*). Using different similarity coefficients such as Dice and simple matching could affect the results of the unweighted pair method of groups with arithmetic (UPGMA) and other methods (*Reif, Melchinger & Frisch, 2005*), whereas Egyptian mango cultivars differ in fruit quality and disease tolerance, but the extent of genetic differences between each of them is not well known. This study aims to evaluate the impact of two DNA markers (RAPD and SSR markers) on the genetic diversity levels of 17 cultivars of mango, as well as to compare two alternative similarity coefficients and clustering techniques to identify the most dissimilar cultivars in genetic diversity assessments that will serve as parents in mango breeding and improvement programs in order to maintain the unique genetic resources.

## MATERIALS AND METHODS

### Plant materials and processing of leaves

All plant materials used and collected in the study met Egypt's guidelines and legislative regulations. Fifty-one mature mango trees (*Mangifera indica* L.) encompassing 17 cultivars were included in the present investigation (Table 1). The experimental trees were grown in private orchards in Abou Swear City, Ismailia Governorate, Egypt. It was chosen for the quality of the fruit produced there. Three trees per cultivar were selected; all of them were vegetatively propagated. The trees were labeled in March and April of 2021 at the time of blooming, and leaf material for DNA extraction was gathered at that time. Care was taken in selecting samples to gather only young, tender, healthy leaves. Samples were wrapped in aluminum foil, appropriately labeled, fixed by submersion briefly in liquid nitrogen, and stored at $-80\,°C$ until DNA extraction.

### DNA extraction

The genomic DNA from leaf samples was extracted by a modified CTAB method (*Porebski, Bailey & Baum, 1997*). DNA was quantified by gel electrophoresis, and its quality was verified by the Nano Drop ND-1000 spectrophotometer (GMI, Ramsey, MN, USA). DNA samples were then stored at $4\,°C$. The stock DNA samples of each cultivar were diluted with tris-EDTA and analyzed individually to detect intra-cultivar variations and bulked to detect inter-cultivar variations.

**Table 1  List of 17 Mango cultivars used in the study.**

| No. | Name | Origin | No. | Name | Origin |
|---|---|---|---|---|---|
| 1 | Banarasi Langra | India | 10 | Sukkary white | Selected seedy clone in Egypt |
| 2 | Mabrouka | India | 11 | Mstikawy | Selected seedy clone in Egypt |
| 3 | Ewais | Selected seedy clone in Egypt | 12 | Fajri kalan | India |
| 4 | Taymour | Selected seedy clone in Egypt | 13 | Elwazza neck | Selected seedy clone in Egypt |
| 5 | Hindi Besennara | India | 14 | Keitt | USA |
| 6 | Mesk | Selected seedy clone in Egypt | 15 | Kent | USA |
| 7 | Zebda | Selected seedy clone in Egypt | 16 | Senarry | Selected seedy clone in Egypt |
| 8 | Hindi mlooky | India | 17 | Nabiel | Selected seedy clone in Egypt |
| 9 | Companeit Elsowwa | Selected seedy clone in Egypt | | | |

## SSR amplification

Thirty of the microsatellite markers utilized in this investigation were previously described by *Duval et al. (2005)*, *Honsho et al. (2005)*, *Schnell et al. (2005)* and *Viruel et al. (2005)* (Table 2) and these primers were synthesized by the Oligosystem (Macrogen, Seoul, Korea). The polymerase chain reactions were performed in 10 µl volume containing: 0.2 µM of each forward and reverse primers, 0.2 mM of dNTPs, 1.8 mM MgCl$_2$, 0.05 U Taq polymerase, 1X PCR buffer, and 10 ng of template DNA, according to *Schnell et al. (2005)* with slight modifications. The amplification was done in a thermocycler (Eppendorf MasterCycler Gradient; Eppendorf, Hamburg, Germany). After a first denaturation step at 94 °C for 2 min, thereafter 30 cycles at 94 °C for 1 min, 51 °C for 30 s, 72 °C for 1 min, and finally extension at 72 °C for 5 min.

Every reaction was repeated twice to ensure the reproducibility of the results. The PCR mixer and cycling PCR products were separated on agarose gel (2%), and ethidium bromide was utilized for staining to ensure the PCR amplification and determine the approximate size of the amplified fragments. Then, products were separated on polyacrylamide gels (7%) to confirm the allele sizing of the SSR loci and stained with ethidium bromide solution and visualized using the gel documentation model (Gel-Doc 2000 with Diversity Database software Ver. 2.1; Bio-Rad Laboratories, Hercules, CA, USA) for gel analysis. Quantity One software was used to estimate the sizes of the products by comparison with the size marker.

## RAPD amplification

Thirty 10-mer RAPD primers were screened for the RAPD-PCR reactions that resulted in distinct and well-separated bands on a polyacrylamide gel. These primers (Table 3) were synthesized by Oligo (Macrogen, Seoul, Korea) and utilized to detect polymorphisms in the 17 mango cultivars. The RAPD-PCR reaction was achieved, according to *Williams et al. (1990)*. PCR amplification was carried out in a total volume of 10 µl, containing 10 ng genomic DNA, 0.3 pmol 10 mer random primer, 1X reaction buffer, 5 µl of Go-Taq (ready mastermix) including (1.5 mM Mg Cl$_2$, 0.2 mM dNTPs, 0.5 U Taq polymerase) PCR amplification was performed in a primus 384 well thermocycler (MWG Biotech AG), Ebensburg, Germany; http://www.mwg-biotech.com), programmed to include a pre-denaturation step at 94 °C for 60 s and followed by 34 cycles of denaturation at 94 °C
**Table 2  List of polymorphic SSR markers used in this study.**

| No. | Locus | SSR primers sequence 5—>>3 | No. | Locus | SSR primers sequence 5—>>3 |
|---|---|---|---|---|---|
| 1 | MiSHRS_1 | F: TAACAGCTTTGCTTGCCTCC<br>R: TCCGCCGATAAACATCAGAC | 16 | mMiCIR_5 | F: GCCCTTGCATAAGTTG<br>R: TAAGTGATGCTGCTGGT |
| 2 | MiSHRS_4 | F: CCACGAATATCAACTGCTGCC<br>R: TCTGACACTGCTCTTCCACC | 17 | mMiCIR_8 | F: GACCCAACAAATCCAA<br>R: ACTGTGCAAACCAAAAG |
| 3 | MiSHRS_18 | F: AAACGAGGAAACAGAGCAC<br>R: CAAGTACCTGCTGCAACTAG | 18 | mMiCIR_9 | F: AAAGATAAGATTGGGAAGAG<br>R: CGTAAGAAGAGCAAAGGT |
| 4 | MiSHRS_32 | R: AGAAACATGATGTGAACC<br>F: TTGATGCAACTTTCTGCC | 19 | mMiCIR_18 | F: CCTCAATCTCACTCAACA<br>R: ACCCCACAATCAAACTAC |
| 5 | mMiCIR_36 | F: GTTTTCATTCTCAAAATGTGTG<br>R: CTTTCATGTTCATAGATGCAA | 20 | mMiCIR_21 | F: CCATTCTCCATCCAAA<br>R: TGCATAGCAGAAAGAAGA |
| 6 | MiSHRS_48 | F: TTTACCAAGCTAGGGTCA<br>R: CACTCTTAAACTATTCAACCA | 21 | mMiCIR_22 | F: TGTCTACCATCAAGTTCG<br>R: GCTGTTGTTGCTTTACTG |
| 7 | LMMA_1 | F:ATGGAGACTAGAATGTACAGAG<br>R: ATTAAATCTCGTCCACAAGT | 22 | mMiCIR_25 | F: ATCCCCAGTAGCTTTGT<br>R: TGAGAGTTGGCAGTGTT |
| 8 | LMMA_7 | F: ATTTAACTCTTCAACTTTCAAC<br>R: AGATTTAGTTTTGATTATGGAG | 23 | MiSHRS_29 | F: CAACTTGGCAACATAGAC<br>R: ATACAGGAATCCAGCTTC |
| 9 | LMMA_8 | F: CATGGAGTTGTGATACCTAC<br>R: CAGAGTTAGCCATATAGAGTG | 24 | mMiCIR_29 | F: GCGTGTCAATCTAGTGG<br>R: GCTTTGGTAAAAGGATAAG |
| 10 | LMMA_9 | F:TTGCAACTGATAACAAATATAG<br>R: TTCACATGACAGATATACACTT | 25 | mMiCIR_30 | F: GCTCTTTCCTTGACCTT<br>R: TCAAAATCGTGTCATTTC |
| 11 | LMMA_10 | F: TTCTTTAGACTAAGAGCACATT<br>R: AGTTACAGATCTTCTCCAATT | 26 | MIAC_2 | F: GCTTTATCCACATCAATATCC<br>R: TCCTACAATAACTTGCC |
| 12 | LMMA_11 | F: ATTATTTACCCTACAGAGTGC<br>R: GTATTATCGGTAATGTCTTCAT | 27 | MIAC_3 | F: TAAGCTAAAAAGGTTATAG<br>R: CCATAGGTGAATGTAGAGAG |
| 13 | LMMA_13 | F: CACAGCTCAATAAACTCTATG<br>R: CATTATCCCTAATCTAATCATC | 28 | MIAC_4 | F: CGTCATCCTTTACAGCGAACT<br>R: CATCTTTGATCATCCGAAAC |
| 14 | LMMA_14 | F: ATTATCCCTATAATGCCCTAT<br>R: CTCGGTTAACCTTTGACTAC | 29 | MIAC_5 | F: AATTATCCTATCCCTCGTATC<br>R: AGAAACATGATGTGAACC |
| 15 | LMMA_15 | F: AACTACTGTGGCTGACATAT<br>R: CTGATTAACATAATGACCATCT | 30 | MIAC_6 | F: CGCTCTGTGAGAATCAAATGGT<br>R: GGACTCTTATTAGCCAATGGGATG |

for 60 s, annealing at 36 °C for 45 s and extension at 72 °C for 30 s, finished with a final extension step of 5 min at 72 °C.

Electrophoresis of the fragments was done on 7% polyacrylamide gels in a vertical CBS Scientific (San Diego, CA, USA; http://www.cbsscientific.com) electrophoresis unit in 1x TBE buffer at a voltage of 480 V and 80 mA for 90 min (until the lower band of dye escaped from the gel). Gel staining was done using silver nitrate according to a modified protocol of *Bassam & Caetano-Anollés (1993)*. Dried gels were scanned, and the sizes of the amplified products were visually examined and estimated by Quantity One software.

## Data scoring and analysis

Bands were scored from the images. The presence of a band was scored as 1, and the absence of a band as 0. Cluster analysis was used to identify the studied populations and to
**Table 3  List of polymorphic RAPD markers used in this study.**

| No. | Name | Sequence | No. | Name | Sequence |
|---|---|---|---|---|---|
| 1 | OPM_20 | 5′-AGG TCT TGG G-3′ | 16 | OPH_18 | 5′-GAA TCG GCC A-3′ |
| 2 | OPQ_05 | 5′-CCG CGT CTT G-3′ | 17 | OPB_17 | 5′-AGG GAA CGA G-3′ |
| 3 | OPS_17 | 5′-TGG GGA CCA C-3′ | 18 | OPA_04 | 5′-AAT CGG GCT G-3′ |
| 4 | OPD_20 | 5′-ACC CGG TCA C-3′ | 19 | OPE_12 | 5′-TTA TCG CCC C-3′ |
| 5 | OPO_12 | 5′-CAG TGC TGT G-3′ | 20 | OPA_01 | 5′-CAG GCC CTT C-3′ |
| 6 | OPA_13 | 5′-CAG CAC CCA C-3′ | 21 | OPX_08 | 5′-CAG GGG TGG A-3′ |
| 7 | OPB_11 | 5′-GTA GAC CCG T-3′ | 22 | OPX_17 | 5′-GAC ACG GAC C-3′ |
| 8 | OPJ_20 | 5′-AAG CGG CCT C-3′ | 23 | OPX_02 | 5′-TTC CGC CAC C-3′ |
| 9 | OPM_06 | 5′-CTG GGC AAC T-3′ | 24 | OPX_01 | 5′-CTG GGC ACG A-3′ |
| 10 | OPA_09 | 5′-GGG TAA CGC C-3′ | 25 | OPA_18 | 5′-AGG TGA CCG T-3′ |
| 11 | OPX-18 | 5′-GAC TAG GTG G-3′ | 26 | OPA_07 | 5′-GAA ACG GGT G-3′ |
| 12 | OPM-12 | 5′-GGG ACG TTG G-3′ | 27 | OPA_11 | 5′-CAA TCG CCG T-3′ |
| 13 | OPD_13 | 5′-GGG GTG ACG A-3′ | 28 | OPW_11 | 5′-CTG ATG CGT G-3′ |
| 14 | OPM_15 | 5′-GAC CTA CCA C-3′ | 29 | OPC_05 | 5′-GAT GAC CGC C-3′ |
| 15 | OPG_11 | 5′-TGC CCG TCG T-3′ | 30 | OPK_10 | 5′-GTG CAA CGT G-3′ |

group similar populations in terms of all measured traits. Various cluster analysis methods and different distance criteria were evaluated, and the method with the highest cophenetic correlation coefficient was selected. The number of amplified bands, polymorphic bands, and polymorphisms percentage for each primer were obtained based on the band patterns and used for grouping the cultivars and estimating the indices.

The polymorphism information content (PIC) (*Botstein et al., 1980*), effective multiplex ratio, mean heterozygosity, marker index (*Powell et al., 1996*), expected heterozygosity (*Liu, 1998*), and discriminating power (*Tessier et al., 1999*) were estimated using an Excel program based on the relevant formulas. The number of effective alleles, Shannon's information index, and Nei's measure of gene diversity were calculated using POPGEN software version 1.31 (*Yeh, 1999*).

The probability of identity (PI) was calculated for each marker according to *Peakall & Smouse (2012)*. Differences and similarities were first calculated based on different similarity criteria for grouping the genotypes cultivars. Then the cultivars were grouped using different cluster analysis methods (*e.g.*, unweighted pair-group method using arithmetic average (UPGMA) and Ward). The cluster analysis method that had the highest cophenetic correlation coefficient was selected. The dendrogram and cluster analysis was performed using NTSYS-PC (numerical taxonomy system) version 2.11 (*Rohlf, 2000*). Structure software (*Pritchard, Stephens & Donnelly, 2000*) was used to identify the population structure in the studied mango cultivar with RAPD and SSR markers data. To choose the optimal level of K, for each K, the first 10,000 repetitions were followed by 10,000 Marcov chain Monte Carlo calculations (MCMCs) based on the mixed model from K equals 2 to K equals 10 with three (3) repetitions. Then the best K utilizing structure harvester software (*Earl & von Holdt, 2012*) was identified based on the $\Delta k$ method (*Evanno, Regnaut & Goudet, 2005*).

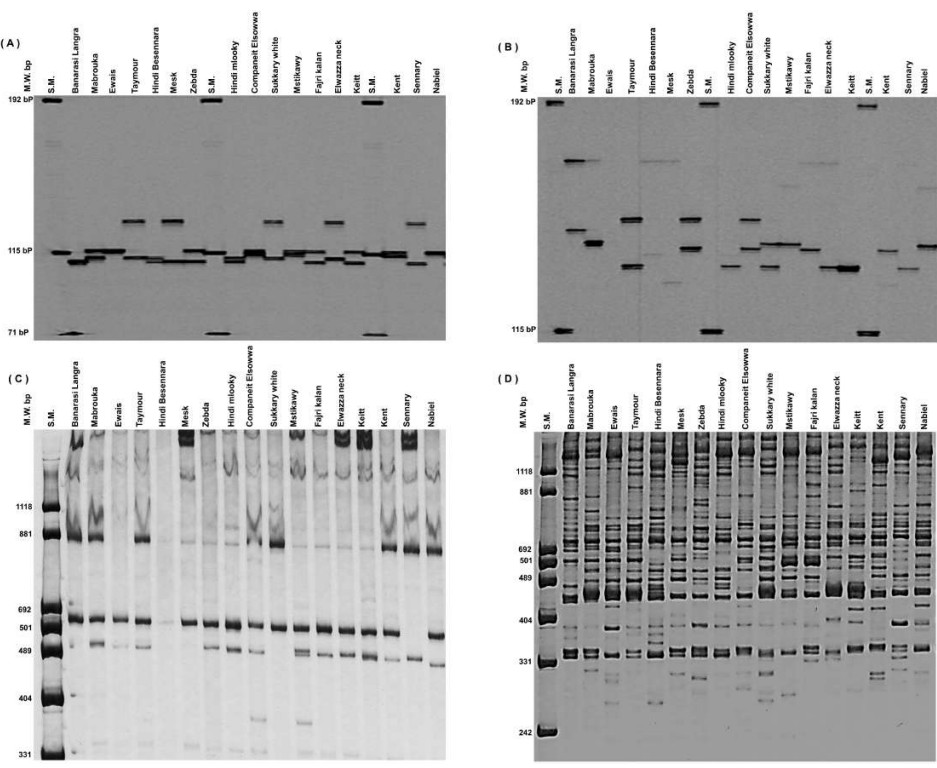

**Figure 1  An example of SSR markers polymorphism.** Miac_4 (A), Miac_5 (B) and RAPD markers OPK_10 (C) and OPA_18 (D) among seventeen mango cultivars.

## RESULTS

### Molecular characterization of SSR and RAPD markers

Thirty RAPD markers (all with multiple loci) produced 434 bands ranging from 6 to 26, with an average of 14.5 bands. OPA_18 had the most bands, with 26, while OPK_10, OPX_02, and OPO_12 had the fewest, with 6 bands, followed by OPG_11 with 8 bands (Figs. 1C and 1D). Thirty RAPD markers generated 361 polymorphic bands (83% polymorphisms), and 73 monomorphic bands, which were typical of all cultivars, existed. The 30 SSR markers (all of them having a single locus) showed high levels of polymorphisms (~100%) and produced 192 polymorphic alleles averaging 6.4/locus. The number of total alleles/loci ranged from 4 (MIAC_4, MiSHRS_18, and mMiCIR_36) to 10 (MIAC_5) (Figs. 1A and 1B).

According to this study's RAPD markers, the expected heterozygosity (He) ranged from 0.415 (OPA_01) to 0.500 (OPQ_05, OPM_06, and OPM_12), with a mean of 0.480. For SSR markers, expected heterozygosity (He) values varied from 0.524 (mMiCIR_21) to 0.0.860 (MIAC_5), with a mean of 0.752.

The PIC value was calculated separately for each studied marker, and the results are shown for the RAPD markers in Table 4 and SSR markers in Table 5. The PIC values, which reflect allele diversity and frequency among the cultivars, were not uniformly higher for

**Table 4 Characteristics and studied indices of RAPD primers used in the study of genetic diversity of mango genotypes.**

| Primer | Na | P | He | PIC | Havp | EMR | MI | D | NEA | H | S |
|--------|-----|------|-------|-------|-------|--------|-------|-------|-------|-------|-------|
| OPM_20 | 21 | 15 | 0.477 | 0.380 | 0.002 | 5.882 | 0.011 | 0.847 | 1.488 | 0.295 | 0.454 |
| OPQ_05 | 24 | 22 | 0.500 | 0.368 | 0.001 | 11.059 | 0.015 | 0.748 | 1.634 | 0.359 | 0.532 |
| OPS_17 | 19 | 15 | 0.484 | 0.376 | 0.002 | 6.176 | 0.012 | 0.831 | 1.473 | 0.290 | 0.450 |
| OPD_20 | 19 | 13 | 0.477 | 0.379 | 0.002 | 5.118 | 0.011 | 0.846 | 1.494 | 0.281 | 0.428 |
| OPO_12 | 6 | 5 | 0.494 | 0.371 | 0.006 | 2.235 | 0.013 | 0.803 | 1.603 | 0.368 | 0.553 |
| OPA_13 | 11 | 7 | 0.470 | 0.383 | 0.004 | 2.647 | 0.010 | 0.859 | 1.403 | 0.264 | 0.420 |
| OPB_11 | 10 | 10 | 0.496 | 0.371 | 0.003 | 4.529 | 0.013 | 0.796 | 1.534 | 0.304 | 0.513 |
| OPJ_20 | 14 | 12 | 0.500 | 0.369 | 0.002 | 5.824 | 0.014 | 0.766 | 1.583 | 0.347 | 0.524 |
| OPA_09 | 13 | 12 | 0.481 | 0.378 | 0.002 | 4.824 | 0.011 | 0.840 | 1.484 | 0.303 | 0.470 |
| OPX_18 | 14 | 14 | 0.487 | 0.375 | 0.002 | 8.118 | 0.017 | 0.665 | 1.527 | 0.328 | 0.501 |
| OPM_06 | 16 | 12 | 0.500 | 0.369 | 0.002 | 6.176 | 0.015 | 0.736 | 1.578 | 0.341 | 0.512 |
| OPM_12 | 13 | 9 | 0.500 | 0.368 | 0.003 | 4.471 | 0.015 | 0.755 | 1.643 | 0.377 | 0.560 |
| OPD_13 | 18 | 13 | 0.486 | 0.375 | 0.002 | 7.000 | 0.017 | 0.661 | 1.605 | 0.356 | 0.532 |
| OPM_15 | 14 | 14 | 0.496 | 0.370 | 0.002 | 6.412 | 0.013 | 0.791 | 1.600 | 0.347 | 0.518 |
| OPG_11 | 8 | 7 | 0.484 | 0.376 | 0.004 | 4.118 | 0.017 | 0.656 | 1.598 | 0.338 | 0.501 |
| OPH_18 | 19 | 18 | 0.490 | 0.373 | 0.002 | 7.706 | 0.012 | 0.818 | 1.517 | 0.312 | 0.478 |
| OPB_17 | 9 | 6 | 0.457 | 0.389 | 0.004 | 2.118 | 0.009 | 0.878 | 1.455 | 0.285 | 0.445 |
| OPA_04 | 16 | 12 | 0.477 | 0.380 | 0.002 | 4.706 | 0.011 | 0.847 | 1.479 | 0.287 | 0.441 |
| OPE_12 | 17 | 14 | 0.489 | 0.374 | 0.002 | 5.941 | 0.012 | 0.821 | 1.469 | 0.297 | 0.460 |
| OPA_01 | 10 | 8 | 0.415 | 0.407 | 0.003 | 5.647 | 0.017 | 0.503 | 1.693 | 0.386 | 0.562 |
| OPX_08 | 17 | 12 | 0.499 | 0.369 | 0.002 | 5.765 | 0.014 | 0.770 | 1.637 | 0.369 | 0.547 |
| OPX_17 | 16 | 13 | 0.491 | 0.373 | 0.002 | 5.647 | 0.013 | 0.812 | 1.472 | 0.287 | 0.447 |
| OPX_02 | 6 | 2 | 0.484 | 0.376 | 0.014 | 0.824 | 0.012 | 0.838 | 1.567 | 0.325 | 0.492 |
| OPX_01 | 10 | 10 | 0.457 | 0.389 | 0.003 | 3.529 | 0.009 | 0.877 | 1.359 | 0.232 | 0.373 |
| OPA_18 | 26 | 22 | 0.420 | 0.405 | 0.001 | 6.588 | 0.007 | 0.911 | 1.392 | 0.248 | 0.393 |
| OPA_07 | 14 | 14 | 0.480 | 0.378 | 0.002 | 5.588 | 0.011 | 0.842 | 1.456 | 0.283 | 0.441 |
| OPA_11 | 17 | 16 | 0.457 | 0.389 | 0.002 | 5.647 | 0.009 | 0.876 | 1.439 | 0.275 | 0.431 |
| OPW_11 | 20 | 20 | 0.497 | 0.370 | 0.001 | 9.176 | 0.013 | 0.790 | 1.588 | 0.352 | 0.529 |
| OPC_05 | 11 | 9 | 0.479 | 0.378 | 0.003 | 3.588 | 0.011 | 0.843 | 1.451 | 0.272 | 0.423 |
| OPK_10 | 6 | 6 | 0.491 | 0.373 | 0.005 | 2.588 | 0.012 | 0.816 | 1.518 | 0.318 | 0.488 |
| Average | 14.5 | 12.03 | 0.480 | 0.378 | 0.003 | 5.322 | 0.013 | 0.795 | 1.525 | 0.314 | 0.481 |
| polymorphism 83% | | | | | | | | | | | |

**Notes.**

NA, Total bands; P, polymorphic bands; He, Expected heterozygosity; PIC, Polymorphism information content; EMR, Effective multiplex ratio; Havp, Mean heterozygosity; MI, Marker index; D, Discriminating power; NEA, Number of effective alleles; H, Nei's gene diversity; S, Shannon's Information index.

all the RAPD loci tested. The PIC value ranged from 0.368 (OPQ_05 & OPM_12) to 0.405 (OPA_18), with a mean of 0.378. On the other hand, PIC values for SSR markers ranged from 0.510 (mMiCIR_21) to 0.852 (MIAC_5), with a mean of 0.735.

The mean heterozygosity (Havp) of the RAPD markers ranged from 0.001 to 0.014 with an average of 0.003; for the SSR markers, it ranged from 0.014 to 0.095 with an average of 0.043. The OPX_02 (RAPD) and the mMiCIR_36 (SSR) had the highest mean heterozygosity value.

**Table 5   Characteristics and studied indices of SSR primers used in the study of genetic diversity of mango cultivars.**

| Primer | NA | P | He | PIC | Havp | EMR | MI | D | NEA | H | S |
|---|---|---|---|---|---|---|---|---|---|---|---|
| MiSHRS_1 | 5 | 5 | 0.767 | 0.748 | 0.047 | 1.412 | 0.066 | 0.815 | 1.361 | 0.254 | 0.414 |
| MiSHRS_4 | 5 | 5 | 0.720 | 0.700 | 0.056 | 1.765 | 0.099 | 0.765 | 1.448 | 0.282 | 0.446 |
| MiSHRS_18 | 4 | 4 | 0.681 | 0.649 | 0.080 | 1.647 | 0.131 | 0.724 | 1.580 | 0.334 | 0.495 |
| MiSHRS_32 | 7 | 7 | 0.749 | 0.733 | 0.036 | 1.706 | 0.061 | 0.796 | 1.311 | 0.204 | 0.338 |
| mMiCIR _36 | 4 | 4 | 0.618 | 0.590 | 0.095 | 1.529 | 0.146 | 0.657 | 1.460 | 0.282 | 0.437 |
| MiSHRS_48 | 5 | 5 | 0.745 | 0.724 | 0.051 | 1.176 | 0.060 | 0.792 | 1.295 | 0.214 | 0.361 |
| LMMA_1 | 9 | 9 | 0.829 | 0.818 | 0.019 | 1.765 | 0.034 | 0.881 | 1.246 | 0.178 | 0.305 |
| LMMA_7 | 7 | 7 | 0.785 | 0.770 | 0.031 | 1.882 | 0.058 | 0.834 | 1.351 | 0.231 | 0.374 |
| LMMA_8 | 8 | 8 | 0.770 | 0.753 | 0.029 | 1.941 | 0.056 | 0.818 | 1.322 | 0.202 | 0.332 |
| LMMA_9 | 5 | 5 | 0.749 | 0.729 | 0.050 | 1.471 | 0.074 | 0.796 | 1.384 | 0.258 | 0.418 |
| LMMA_10 | 8 | 8 | 0.782 | 0.768 | 0.027 | 2.000 | 0.054 | 0.831 | 1.324 | 0.209 | 0.344 |
| LMMA_11 | 6 | 6 | 0.807 | 0.793 | 0.032 | 1.765 | 0.057 | 0.857 | 1.379 | 0.263 | 0.426 |
| LMMA_13 | 5 | 5 | 0.749 | 0.727 | 0.050 | 1.529 | 0.077 | 0.795 | 1.404 | 0.268 | 0.425 |
| LMMA_14 | 7 | 7 | 0.793 | 0.779 | 0.030 | 1.765 | 0.052 | 0.843 | 1.328 | 0.222 | 0.366 |
| LMMA_15 | 6 | 6 | 0.719 | 0.696 | 0.047 | 1.882 | 0.088 | 0.764 | 1.410 | 0.247 | 0.386 |
| mMiCIR_5 | 6 | 6 | 0.751 | 0.730 | 0.041 | 1.824 | 0.076 | 0.798 | 1.415 | 0.256 | 0.400 |
| mMiCIR_8 | 8 | 8 | 0.839 | 0.829 | 0.020 | 1.941 | 0.039 | 0.892 | 1.306 | 0.219 | 0.366 |
| mMiCIR_9 | 6 | 6 | 0.793 | 0.777 | 0.021 | 3.039 | 0.063 | 0.952 | 1.399 | 0.267 | 0.371 |
| mMiCIR_18 | 9 | 9 | 0.838 | 0.829 | 0.018 | 1.471 | 0.026 | 0.891 | 1.196 | 0.152 | 0.274 |
| mMiCIR_21 | 5 | 5 | 0.524 | 0.510 | 0.095 | 0.882 | 0.084 | 0.557 | 1.233 | 0.150 | 0.257 |
| mMiCIR_22 | 7 | 7 | 0.814 | 0.801 | 0.027 | 1.647 | 0.044 | 0.865 | 1.296 | 0.213 | 0.358 |
| mMiCIR_25 | 5 | 5 | 0.675 | 0.655 | 0.065 | 1.176 | 0.076 | 0.717 | 1.305 | 0.204 | 0.340 |
| MiSHRS _29 | 5 | 5 | 0.584 | 0.561 | 0.083 | 1.588 | 0.132 | 0.621 | 1.273 | 0.193 | 0.325 |
| mMiCIR_29 | 9 | 9 | 0.827 | 0.817 | 0.019 | 1.765 | 0.034 | 0.878 | 1.245 | 0.177 | 0.305 |
| mMiCIR_30 | 6 | 6 | 0.782 | 0.766 | 0.036 | 1.765 | 0.064 | 0.831 | 1.385 | 0.256 | 0.411 |
| MIAC_2 | 5 | 5 | 0.692 | 0.666 | 0.062 | 1.353 | 0.083 | 0.735 | 1.360 | 0.232 | 0.371 |
| MIAC_3 | 7 | 7 | 0.752 | 0.733 | 0.035 | 1.294 | 0.046 | 0.799 | 1.235 | 0.166 | 0.285 |
| MIAC_4 | 4 | 4 | 0.737 | 0.713 | 0.053 | 1.529 | 0.081 | 0.783 | 1.511 | 0.332 | 0.513 |
| MIAC_5 | 10 | 10 | 0.860 | 0.852 | 0.014 | 1.647 | 0.023 | 0.913 | 1.197 | 0.154 | 0.276 |
| MIAC_6 | 9 | 9 | 0.831 | 0.821 | 0.019 | 1.765 | 0.033 | 0.883 | 1.245 | 0.178 | 0.306 |
| Average | 6.4 | 6.4 | 0.752 | 0.735 | 0.043 | 1.664 | 0.067 | 0.803 | 1.340 | 0.227 | 0.369 |
| Polymorphism 100% | | | | | | | | | | | |

**Notes.**

NA, Total bands;  P,  polymorphic bands; He, Expected heterozygosity; PIC, Polymorphism information content; EMR, Effective multiplex ratio; Havp, Mean heterozygosity; MI, Marker index; D, Discriminating power; NEA, Number of effective alleles; H, Nei's gene diversity; S, Shannon's Information index.

The effective multiple ratio (EMR), which indicates the number of polymorphic gene loci in germplasm, ranged from 0.824 for OPX_02 to 11.059 for OPQ_05 of the RAPD markers with an average of 5.322, while for the SSR markers, it ranged from 0.882 to 3.039 for the MiCIR_21 and mMiCIR_9, respectively, with an average of 1.664 (Tables 4 and 5).

The marker index (MI) was developed to evaluate how well the markers could detect polymorphisms. The highest value of the RAPD marker index was 0.017 for the following four markers: OPX_18, OPD_13, OPG_11, and OPA_01, while the lowest value was 0.007 for OPA_18 and an average of 0.013 (Table 4). The SSR markers had an average marker

index of 0.067, with the MIAC_5 having the lowest value (0.023) and the MiCIR_36 having the highest value (0.146) (Table 5).

Discrimination power varied from 0.503 for OPA_01 to 0.911 for OPA_18 with an average of 0.795 for the RAPD markers, and for the SSR markers, it varied from 0.557 to 0.952 for MiCIR_21 and mMiCIR_9, respectively with an average of 0.

The number of effective alleles for the RAPD markers ranged from 1.359 (OPX_01) to 1.693(OPA_01), and the mean in the study population was 1.525 Table 4). On the other hand, it ranged from 1.196 (mMiCiR_18) to 1.58 (MiSHRS_18), and its mean in the study population was 1.34 for the SSR markers (Table 5).

The Nei gene diversity index (H) is one of the most crucial metrics for assessing gene diversity among populations and cultivars. The RAPD markers' H values range from 0.232 to 0.386 and an average of 0.314. The OPX_01 displayed the lowest amount of H, and the OPA_01 displayed the greatest value of H (Table 4). SSR markers had an average value of H (0.226), with mMiCIR_21 recording the lowest value (0.150) and MiSHRS_18 recording the highest value (0.334). The Shannon index shows the level of variation between cultivars. The mean Shannon index of RAPD markers in this study was 0.481.

OPA_01 had the highest value (0.562), followed by OPM_12 and OPO_12 (0.560–0.553). The OPX_01 value (0.373) was the lowest. The mean Shannon index for SSR markers was 0.369. The MIAC_4 and MiSHRS_18 showed the highest values (0.513 and 0.495. respectively), while the mMiCIR_21 had the lowest value (0.257).

## Power of the probability of identity

According to *Mirimin et al. (2015)*, the probability of identify (PI) represents the possibility of discovering two people with the same genotype at particular loci in the population. The SSR probability value ranged from 0.475 (mMiCIR_21) to 0.14 (MIAC_5), and was typically low, with an average of 0.25. The RAPD markers' probability value was very low and ranged from 0.224 (OPK_10) to 0.053 (OPQ_05), with a mean of 0.109 (Table 6).

## Genetic similarity and cluster analysis

The pairwise comparison of cultivars based on simple matching similarity coefficients indicated likely high genetic similarity between the 17 mango cultivars, ranging from a maximum of 0.83 for 'Hindi Besennara' and 'Hindi mlooky' to a minimum of 0.58 for 'Zebda' and 'Elwazza neck' for the SSR markers and from a maximum of 0.80 for the 'Taymour', 'Elwazza neck', and 'Sukkary white' to a minimum of 0.65 for the 'Keitt' and 'Ewais' for the RAPD markers. A dendrogram was generated from the binary data of the SSR marker score results based on the simple matching similarity coefficients, as shown in Fig. 2A. The dendrogram showed that the genetic similarity coefficient of the 17 mango cultivars ranged from 0.58 to 0.83. It can be seen that the 17 mango cultivars were divided into three clusters, with a mean similarity of 0.66 for cluster 1 (11 cultivars) and cluster 2 (5 cultivars). The third cluster contained only the 'Zebda' cultivar. The first cluster was divided into two groups with a mean of 0.67. Group 1 had two cultivars, 'Banarasi Langra' and 'Fajri kalan', with a similarity coefficient 0.72. Group 2 was divided into two subgroups, A and B, with a similarity coefficient of 0.68. Subgroup A had seven cultivars, and Subgroup

**Table 6  Values of probability of identify for SSR and RAPD markers.**

| No. | SSR markers | Probability of identify | No. | RAPD markers | Probability of identify |
|---|---|---|---|---|---|
| 1 | MiSHRS_1 | 0.233 | 1 | OPM_20 | 0.066 |
| 2 | MiSHRS_4 | 0.280 | 2 | OPQ_05 | 0.053 |
| 3 | MiSHRS_18 | 0.318 | 3 | OPS_17 | 0.08 |
| 4 | MiSHRS_32 | 0.250 | 4 | OPD_20 | 0.075 |
| 5 | mMiCIR _36 | 0.381 | 5 | OPO_12 | 0.199 |
| 6 | MiSHRS_48 | 0.255 | 6 | OPA_13 | 0.126 |
| 7 | LMMA_1 | 0.171 | 7 | OPB_11 | 0.149 |
| 8 | LMMA_7 | 0.214 | 8 | OPJ_20 | 0.091 |
| 9 | LMMA_8 | 0.230 | 9 | OPM_06 | 0.079 |
| 10 | LMMA_9 | 0.251 | 10 | OPA_09 | 0.107 |
| 11 | LMMA_10 | 0.217 | 11 | OPX_18 | 0.093 |
| 12 | LMMA_11 | 0.193 | 12 | OPM_12 | 0.092 |
| 13 | LMMA_13 | 0.251 | 13 | OPD_13 | 0.066 |
| 14 | LMMA_14 | 0.206 | 14 | OPM_15 | 0.091 |
| 15 | LMMA_15 | 0.281 | 15 | OPG_11 | 0.161 |
| 16 | mMiCIR_5 | 0.248 | 16 | OPH_18 | 0.073 |
| 17 | mMiCIR_8 | 0.161 | 17 | OPB_17 | 0.154 |
| 18 | mMiCIR_9 | 0.207 | 18 | OPA_04 | 0.089 |
| 19 | mMiCIR_18 | 0.162 | 19 | OPE_12 | 0.082 |
| 20 | mMiCIR_21 | 0.475 | 20 | OPA_01 | 0.113 |
| 21 | mMiCIR_22 | 0.186 | 21 | OPX_08 | 0.072 |
| 22 | mMiCIR_25 | 0.325 | 22 | OPX_17 | 0.091 |
| 23 | MiSHRS _29 | 0.416 | 23 | OPX_02 | 0.191 |
| 24 | mMiCIR_29 | 0.173 | 24 | OPX_01 | 0.182 |
| 25 | mMiCIR_30 | 0.217 | 25 | OPA_18 | 0.063 |
| 26 | MIAC_2 | 0.308 | 26 | OPA_07 | 0.107 |
| 27 | MIAC_3 | 0.248 | 27 | OPA_11 | 0.089 |
| 28 | MIAC_4 | 0.263 | 28 | OPW_11 | 0.060 |
| 29 | MIAC_5 | 0.140 | 29 | OPC_05 | 0.139 |
| 30 | MIAC_6 | 0.168 | 30 | OPK_10 | 0.224 |
| Sum | | 7.428 | Sum | | 3.26 |
| Average | | 0.25 | Average | | 0.109 |

B had two cultivars, Ewais and Nabiel. The second cluster contained five cultivars, and two of the cultivars, 'Hindi Besennara' and 'Hindi mlooky', achieved a similarity coefficient of 0.83.

The dendrogram was generated from the binary data of the RAPD marker scoring results based on similarity coefficients for simple matches, as shown in Fig. 2B. From the dendrogram, it can be seen that the genetic similarity coefficient of the 17 mango cultivars varied from 0.68 to 0.80. In this dendrogram, the 17 mango cultivars were divided into three clusters with a similarity coefficient of 0.69, namely, cluster 1 (12 cultivars), cluster 2 (four cultivars), and cluster 3 (one cultivar). The first cluster was separated into two

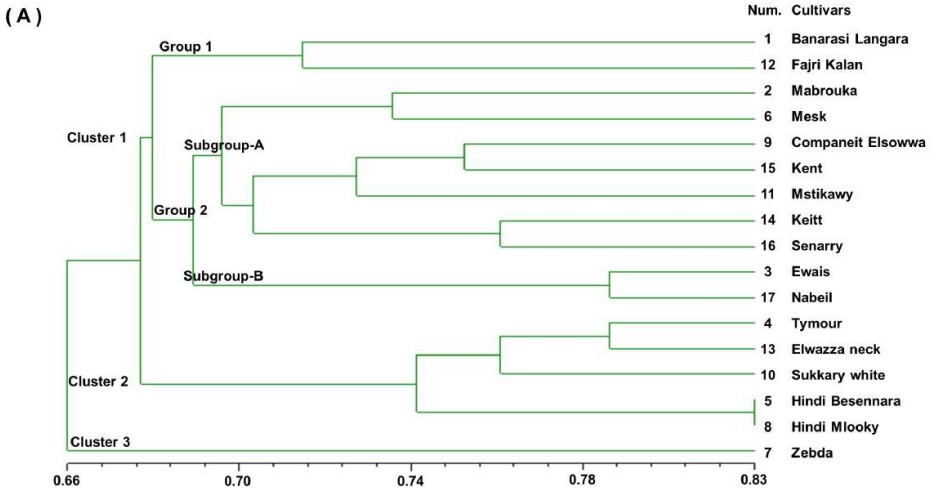

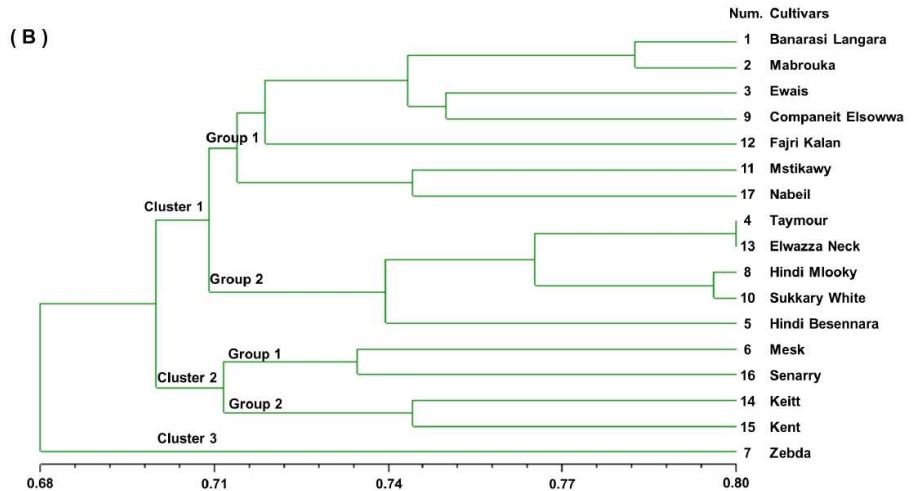

**Figure 2** Genetic similarity dendrogram (SM coefficient) based on unweighted pair-group method using arithmetic average (UPGMA) analysis and constructed of SSR marker (A) and RAPD marker (B) shows the relationships among the 17 mango cultivars.

groups with a similarity coefficient of 0.71. Group1 had seven cultivars, 'Banarasi Langra', 'Mabrouka', 'Ewais', 'Companeit Elsowwa', 'Fajri kalan', 'Mstikawy' and 'Nabiel', with a mean similarity of 0.72.

Group 2 had five cultivars, 'Taymour', 'Elwazza neck', 'Hindi mlooky', 'Sukkary white', and 'Hindi Besennara'. The 'Tayimour' and 'Elwazza neck' cultivars revealed a mean similarity of 0.80. In the second cluster, there were two groups. Group 1 contained 'Mesk' and 'Senarry' and had a coefficient of 0.73. Group 2 consisted of 'Keitt' and 'Kent' and had a similarity coefficient of 0.74. Zebda, the sole cultivar in the third cluster, was the most diverse of the 17 cultivars and appeared as an outlier for both the SSR and RAPD markers.

Dice similarity values were generated for the 30 SSR markers in the 17 mango cultivars (not shown). The Dice pairwise similarity coefficients and similarity values varied from 0.22 between 'Mstikawy' and 'Banarasi Langra' to 0.67 between 'Hindi mlooky' and 'Hindi Besennara', and from a minimum of 0.66 for 'Ewais', 'Sennary', and 'Keitt' to a maximum of 0.81 for 'Elwazza neck' and 'Taymour' for the RAPD markers.

A UPGMA dendrogram was generated using Dice similarity coefficients and applied for SSR markers, as shown in Fig. 3A. The 17 mango cultivars were divided into three main clusters. Cluster 1 contained 'Banarasi Langra' and 'Fajri kalan', with a similarity coefficient of 0.45. Cluster 2 was divided into two subgroups. 'Mabrouka' and 'Mesk' were in group 1. The second group had three subgroups. Subgroup A contained 'Ewais' and 'Nabiel'; subgroup B had 'Companeit Elsowwa', 'Kent', and 'Mstikawy'; and subgroup C had 'Keitt' and 'Senarry'. The 'Zebda' cultivar was the only cultivar in the second group. Cluster 3 consisted of two groups. Group one contained 'Taymour', 'Elwazza neck', and 'Sukkary white'. 'Taymour' and Elwazza neck had similar coefficients of 0.61. 'Hindi mlooky' and 'Hindi Besennara' were tied in the second group, with a coefficient of 0.67.

The dendrogram was generated from the binary data of the RAPD marker scoring results based on the Dice similarity coefficients, as shown in Figs. 3B. The dendrogram shows that the value of the genetic similarity coefficient of the 17 mango cultivars varied from 0.70 to 0.81. The 17 mango cultivars were separated into three clusters with a coefficient of 0.70; they were Cluster 1 (12 cultivars), cluster 2 (four cultivars), and Cluster 3 (one cultivar). The first cluster was divided into three groups with a similarity coefficient of 0.71. Group 1 consisted of six cultivars, 'Banarasi Langra', 'Mabrouka', 'Ewais', 'Companeit Elsowwa', 'Mstikawy', and 'Nabiel', with a similarity coefficient of 0.73. Group 2 had five cultivars, 'Taymour', 'Elwazza neck', 'Hindi mlooky', 'Sukkary white', and 'Hindi Besennara'. The cultivar 'Fajri kalan' was the only one in the third group.

The 'Taymour' and 'Elwazza neck' cultivars had a coefficient of 0.81. The second cluster was divided into two groups. Group 1 contained 'Mesk' and 'Sennary', with a coefficient of 0.73. Group 2 consisted of 'Keitt' and 'Kent', with a mean similarity of 0.75. The third cluster contained 'Zebda'. Notably, 'Zebda' was the most diverse of the 17 cultivars, appearing as an outlier in the UPGMA dendrogram for the SSR and RAPD data.

## Population structure

The population structure was determined based on the model presented in the structure software. Based on the data, the suitable number of groups for the population was two to 10. Using the structure harvester website for the best k, four subgroups were selected for the population (Fig. 4). The cluster analysis was based on the Bayesian statistical model to understand the populations' distance structure. Assuming that the lineage model was of a mixed type and the allelic abundance model was of a continuous type, the results showed that there were four populations of germplasms based on cultivar and genome. These cultivars were not completely separate (Fig. 5), and each cultivar was assigned to each group with almost equal probabilities.

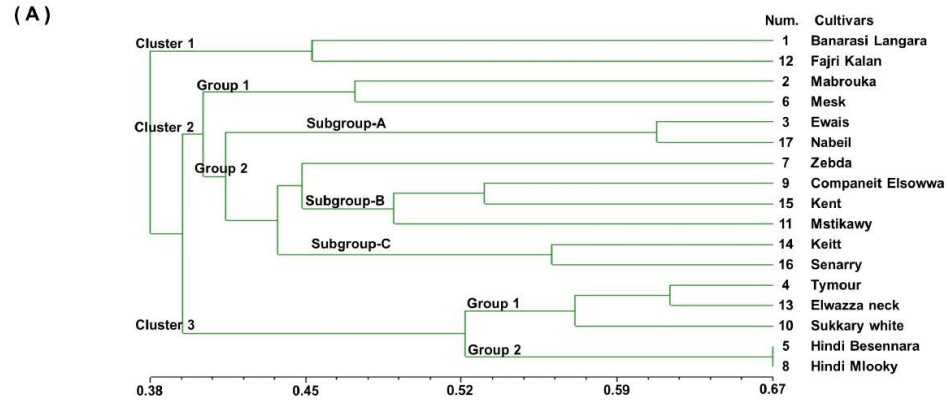

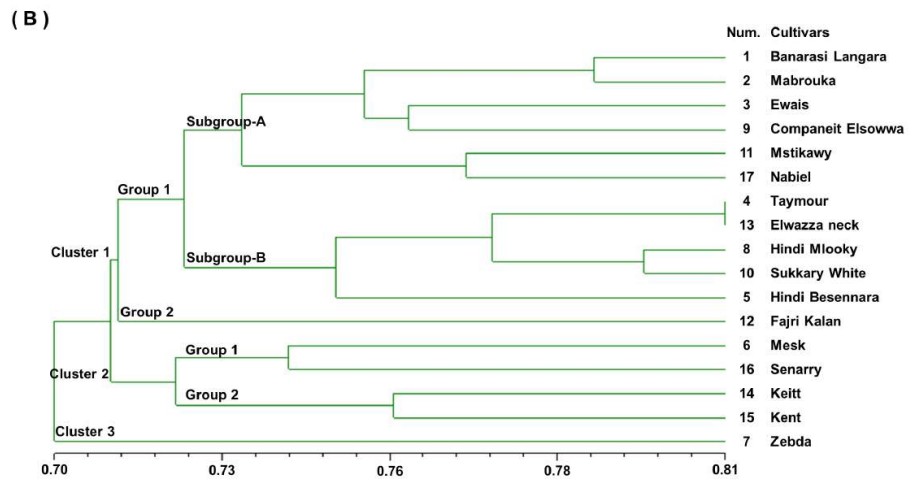

**Figure 3** Genetic similarity dendrogram (Dice coefficient) based on unweighted pair-group method using arithmetic average (UPGMA) analysis and constructed of SSR marker (A) and RAPD marker (B) shows the relationships among the 17 mango cultivars.

## DISCUSSION

The genetic diversity of 17 mango cultivars was investigated in this work using two PCR-based systems: SSR and RAPD. The kind and quantity of polymorphisms identified vary depending on the system in principle. Despite the number of polymorphic alleles generated by 30 SSR markers being different compared to the number of polymorphic bands generated by RAPD markers, they recognized 100% of the polymorphisms. The differences in polymorphisms of the DNA markers generated by each primer indicated the complexity of the plant genome, as the DNA bands resulted from the binding of primer nucleotides in plant chromosomes. However, the number of polymorphic DNA bands was

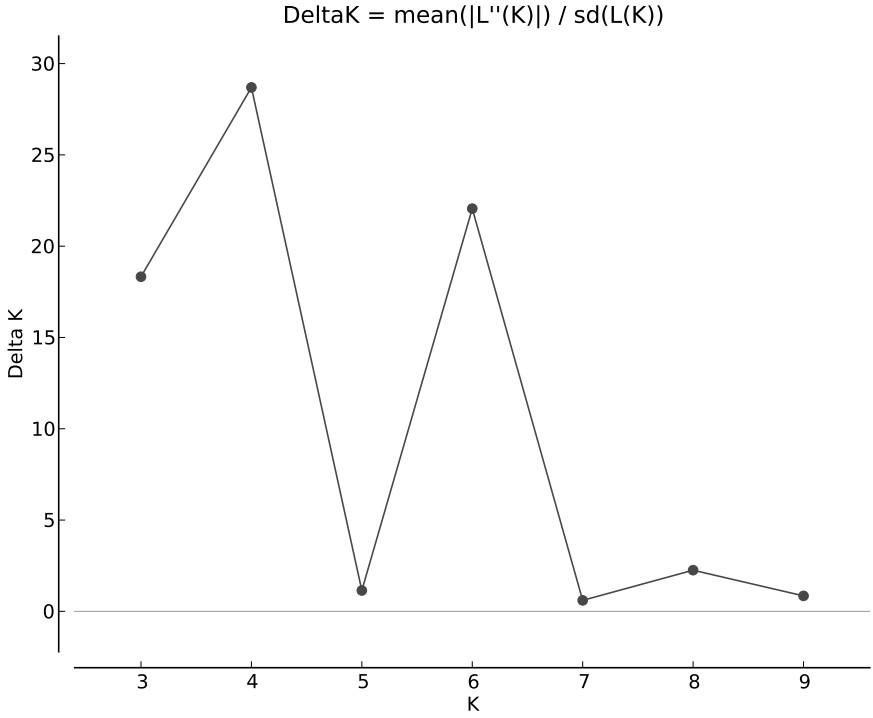

**Figure 4** Determining subgroups using Structure Harvester.

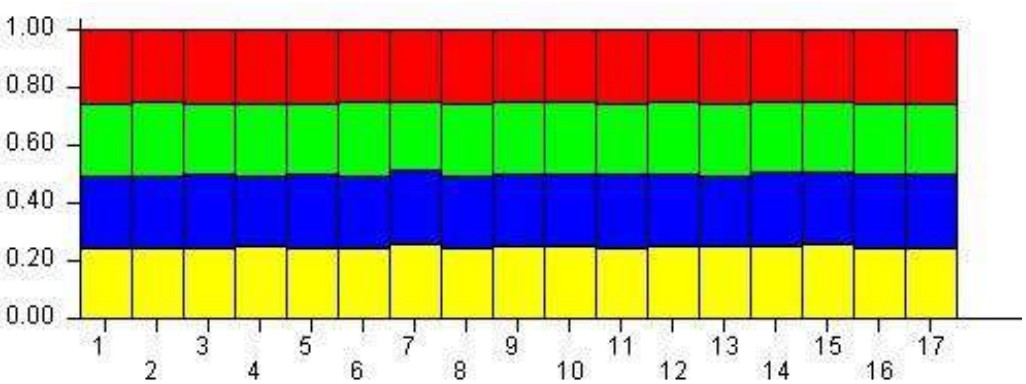

**Figure 5** **The demographic structure of mango cultivars using the Bayesian clustering approach by STRUCTURE software.** The yellow indicates Group 1; Blue is Group 2; green is Group 3, and red is Group 4. The numbers represent genotypes.

able to indicate the genomic profile of the mango species due to the distribution of the primer binding sites (*Fitmawati Hartana & Purwoko, 2010*).

Nevertheless, a substantial percentage of polymorphism from 30 RAPD markers was still present, and this was deemed sufficient and particularly instructive in calculating genetic diversity. Previous studies of mangoes have reported similar levels of polymorphisms

associated with RAPD markers (*Islam et al., 2018*). According to the banding patterns obtained from RAPD and SSR markers, the 17 tested mango cultivars could be distinguished from each other. These results confirmed previous findings that the mango is highly heterozygous (*Sherman, Rubinstein & Eshed, 2015*). This could be explained by the fact that the mating system in mangoes typically involves outcross pollination with some self-pollination, and as a result, the cultivars employed acquired enough free external gene flow (*Kumari et al., 2020*).

The PIC was determined by the quantity of alleles found, their frequency of distribution, the placement of the study's primers in the genome, and the sensitivity of the genotype to the technique employed (*Pachauri et al., 2013*; *Abd El-Moneim et al., 2021*). The 30 RAPD markers utilized in the 17 mango cultivars had a moderate polymorphism, as indicated by the mean of PIC value. This result was greater than the mean showed by *Srivastava et al. (2012)*. However, the average PIC value of RAPD markers studied by various studies differs depending on how many RAPD markers were utilized and how many cultivars were examined. The average PIC value of the present analysis supports previously published mean PIC values for SSR markers in mango (*Ravishankar et al., 2015*; and *Padmakar, Dinesh & Ravishankar, 2017*). Allelic variation may be revealed more readily with a high PIC value. In contrast to RAPD markers, the PIC values for SSR markers were obviously greater.

It is important to note that the high mutational rate at SSR loci, which is impacted by the structure, number of repeated nucleotides, and type of the locus (g-SSR or est-SSR), may potentially be a contributing factor to the level of genetic variability. While most of the amplified SSR loci in the current investigation are based on dinucleotide repeats, loci with a small number of repeated units display a high mutational rate, as *Gadaleta et al. (2007)* described. Because of this, a heterozygous state could be produced by any of the mutated alleles. Consequently, based on the PIC value in this investigation, the SSR markers were more discriminative than the RAPD markers.

Probability of identity (PI) is the probability that two individuals in a population or sample will have the same genotype by chance rather than by relationship (*Mirimin et al., 2015*). The average PI values resulting from SSR markers were small compared to the RAPD markers, which were very small. This indicated that a small number of identical alleles were discovered in the mango cultivars analyzed. These results matched those of *Ravishankar et al. (2015)*, who identified 387 mango accessions using six microsatellite markers. They suggested that the probability of finding two individuals with the same genotype was very close to zero, considering both the marker set and the sample size utilized. However, the PI is considered the most widely used theoretical estimator for accessing supporting statistics for individual identification and quantifying the level of genetic variability in populations or samples (*Dokupilová et al., 2014*; *Mollet et al., 2015*).

Compared to RAPD data, the SSR data in this investigation produced lower similarity values. This was primarily due to the co-dominant character of the SSR markers. Which made it possible to identify many alleles per locus and led to larger levels of anticipated heterozygosity than would be conceivable with RAPD markers. More dinucleotide-type

SSRs were employed than other types, and their mutation rates were considerably higher (*Islam et al., 2018* and *Padmakar, Dinesh & Ravishankar, 2017*).

The SSR data exhibited that the higher the value of the similarity coefficient between the cultivars 'Hindi Besennara' and 'Hindi mlooky' (and they are in the same place in the dendrogram clusters), the more similar the DNA banding patterns would become between the cultivars, meaning that the cultivars were becoming more and more similar. In contrast, the RAPD data showed that 'Taymour' and 'Elwazza neck' were in the same position in the dendrogram clusters. This could not be related to the nature of the used similarity coefficients set but somewhat to a limited number of genomic regions where varieties differed. Such differences could not be assessed with a few markers (*Denčić et al., 2016*).

The selection of the similarity coefficient must be based on several criteria because even a few structural changes in more differentiated groups can change the relationship between varieties with high genetic similarity. Considering the genetic basis of the RAPD markers (*Williams et al., 1990*), the lack of amplification of a particular band in two cultivars did not necessarily represent a genetic similarity between them, so coefficients that exclude these common negative occurrences in their expressions of similarity (*e.g.*, Dice, simple matching) are better suited for use with this type of marker.

Remarkably, all the coefficients showed that the 'Zebda' cultivar was clearly separate from the other mango cultivars in the SSR and RAPD groups. There is no doubt that this cultivar is genetically far from other cultivars, which is essential in breeding and improving mangoes. The reason for this is probably its segregation from the base population in the first selection phase. The dendrograms for the SSR and RAPD markers were not affected by the coefficient type, even by simple matching or Dice. The UPGMA method can provide consistent results in terms of clustering regardless of the similarity coefficient.

Furthermore, the dendrograms generated from the examined coefficients all showed the same general structure (Figs. 2 and 3), so it is evident that the different coefficients caused few changes depending on the type of markers. Nevertheless, different groups were formed for the SSR markers than for the RAPD markers, which is illustrated by the several properties of these markers (*Garcia et al., 2004*). Consequently, the dendrogram for either the SSR markers or RAPD markers in Figs. 2 and 3 obtained by the simple matching similarity coefficient with a single hit was identical to that obtained by Dice.

The 17 cultivars in each of the four subpopulations were almost identical to the other subgroups in other parts of their genome, which is why all mango cultivars were placed in a mixed structure (the probability of each cultivar belonging to each subgroup was less than 0.7). No cultivar was definitely in a specific group. Due to gene flow over time and the possibility of shared genetic ancestry amongst cultivars, this mixing may have occurred. The type of markers used or the fact that the chosen markers were not evenly dispersed across the genome could also be to blame. Additionally, some of these cultivars originated in America and India, while the remainder had been chosen and developed in Egypt. As a result, these chosen cultivars were genetically diverse, most of which were crossbred.

## CONCLUSIONS

The extent of mango polymorphism has been usefully shown by both the RAPD and SSR marker approaches. They are more helpful in evaluating the genetic diversity of the studied cultivars. Although reliability and transferability are two drawbacks of RAPD-based analysis, RAPD data can become highly reliable, provided a set methodology is followed. However, the findings showed that RAPD and SSR markers systems effectively classify the 17 mango cultivars based on where they were first cultivated. Interestingly, it is essential to emphasize that SSRs demonstrated superior performance by displaying higher values for most of the parameters that determine the potential of markers in diversity analysis. Also, it became clear that this investigation's simple matching or Dice similarity coefficients had no bearing on the outcomes. Furthermore, the population structure analysis data will be valuable for conducting association mapping in mango for a cultivar of characteristics.

On the other hand, it became evident that the 'Zebda' cultivar differed genetically from the other mango cultivars evaluated and was further distinct from them. It will probably participate in breeding and enhancement efforts for mangoes in Egypt as a good parent. In particular, considering that it produces fruit of high quality and is resistant to diseases and unfavorable circumstances in the environment. It is important to note that every result from this study will be helpful information for identifying markers for future studies, describing germplasm, breeding, and managing mango germplasm.

### Funding

The authors received support from The Deanship of Scientific Research (DSR) at King Abdulaziz University, Jeddah, Grant No. [IFPRC-187-130-2020]. The funders had no role in study design, data collection and analysis, decision to publish, or preparation of the manuscript.

### Grant Disclosures

The following grant information was disclosed by the authors:
The Deanship of Scientific Research (DSR) at King Abdulaziz University, Jeddah: IFPRC-187-130-2020.

### Competing Interests

Diaa Abd El-Moneim is an Academic Editor for PeerJ.

### Author Contributions

- Mohammed A.A. Hussein conceived and designed the experiments, performed the experiments, analyzed the data, prepared figures and/or tables, authored or reviewed drafts of the article, and approved the final draft.
- Manal Eid conceived and designed the experiments, performed the experiments, analyzed the data, prepared figures and/or tables, authored or reviewed drafts of the article, and approved the final draft.

- Mehdi Rahimi conceived and designed the experiments, performed the experiments, analyzed the data, prepared figures and/or tables, authored or reviewed drafts of the article, and approved the final draft.
- Faten Zubair Filimban conceived and designed the experiments, performed the experiments, analyzed the data, prepared figures and/or tables, authored or reviewed drafts of the article, and approved the final draft.
- Diaa Abd El-Moneim conceived and designed the experiments, performed the experiments, analyzed the data, prepared figures and/or tables, authored or reviewed drafts of the article, and approved the final draft.

## Data Availability

   The raw data is available in the Supplemental Files.

## Supplemental Information

Supplemental information for this article can be found online at http://dx.doi.org/10.7717/peerj.15722#supplemental-information.

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
