# Peer review of "Comparative Assessment of SSR and RAPD markers for genetic diversity in some Mango cultivars"

_PeerJ, doi:10.7717/peerj.15722_

## Round 0.1 · original submission · Major Revisions

Dear Authors
We welcome you to improve and resubmit your manuscript, taking into account the comments of the reviewers. In addition, my personal opinion is that you should detail the method part. In line 174, SCoT markers are mentioned, but you say that you used RAPD and SSR. There is some confusion here. You need to explain and correct this. You may need to redo some calculations. I suggest you check the formulas. Re-evaluate your comment on the effectiveness of RAPD markers. You know, the reliability of the results of RAPD markers is controversial and not reproducible.
Best regards

Reviewer 1 ·

Basic reporting

The manuscript is about studying the genetic diversity of mango cultivars from a molecular point of view and by RAPD and SSR markers. It shows the efficiency of these two markers in the study of genetic diversity of mango, which is useful for breeding programs and conservation of genetic diversity. The manuscript has flaws that I have mentioned and it can be accepted after solving the problems. English throughout the manuscript needs to be improved. Grammatical and writing errors can be seen. Figures and tables are appropriate. The hypotheses are consistent with the reported final results.

Experimental design

Sampling of trees for study was done from several areas in Egipt and due to the fact that the plant is heterozygous, three tree samples of each cultivar were used for DNA extraction.
I recommend that the article be improved in terms of English. The results of statistical analyzes should be checked and more suitable methods should be examined (especially in cluster analysis).

Validity of the findings

The most important issue in your manuscript is correcting some parameters and calculated statistical methods and improving the article from the English point of view. In the entire manuscript, additional explanations should be removed.A stronger discussion should be written.

Additional comments

Dear editor
The manuscript is about studying the genetic diversity of mango cultivars from a molecular point of view and by RAPD and SSR markers. It shows the efficiency of these two markers in the study of genetic diversity of mango, which is useful for breeding programs and conservation of genetic diversity. The manuscript has flaws that I have mentioned and it can be accepted after solving the problems. English throughout the manuscript needs to be improved. Grammatical and writing errors can be seen. Figures and tables are appropriate. The hypotheses are consistent with the reported final results. Sampling of trees for study was done from several areas in Egipt and due to the fact that the plant is heterozygous, three tree samples of each cultivar were used for DNA extraction.


Support criticisms with evidence from the text or from other sources
Line 206: Check the MI calculation method. It seems that the numbers are not correct. Check the formula and source below
MI=EMR*PIC
Varshney RK, Chabane K, Hendre PS, Aggarwal RK, Graner A. Comparative assessment of EST-SSR, EST-SNP and AFLP markers for evaluation of genetic diversity and conservation of genetic resources using wild, cultivated and elite barleys. Plant Sci. 2007;173:638–649. doi: 10.1016/j.plantsci.2007.08.010.
Give specific suggestions on how to improve the manuscript
Line 46-47: What do you mean by this sentence?
The 46 comparison between the RAPD and SSR markers disclosed that the qualitative nature of the data 47 was less for RAPD (0.46) than for SSR (0.75).
Using a similarity coefficient is sufficient. The simple matching coefficient shows the similarity value more than the actual value
Line 56: Keywords listed in alphabetical order
Introduction
The references used in the introduction are old. Bring new references
Material and methods
The concentration of the PCR components is given for SSR and RAPD markers
What was the gel type and concentration for fragment sepration of RAPD markers
Determine whether the mango is homozygous or heterozygous?
Line 174: Is the Scot marker also used?
Line 187: What do you mean by this sentence?
72 monomorphic bands which were conserved in all cultivars.
Line 201:The expected heterozygosity is usually indicated by He. The whole text should be corrected.
Line 218: What is the detection power varied?
Line 233: Nei's genetic diversity index (H) is correct. The whole text should be corrected.
Line 245: Please provide reference.
Line 252 : In my opinion, all analysis based on the simple matching method should be removed.
Line 251: What was the statistical basis for grouping cultivars in cluster analysis?
Line 261: which coefficient do you mean? (a coefficient of 0.66 for 261 cluster 1)
Avoid repeating the results in the discussion section.
Line 442: Considering that mango is cross-pollinating and heterozygous, were heterozygous bands observed in the SSR marker and how was the scoring method?
Figures: All dendrograms show a chaining effect. Have other algorithms been investigated?

Organize by importance of the issues, and number your points
The most important issue in your manuscript is correcting some parameters and calculated statistical methods and improving the article from the English point of view. In the entire manuscript, additional explanations should be removed.
Comment on strengths (as well as weaknesses) of the manuscript
I recommend that the article be improved in terms of English. The results of statistical analyzes should be checked and more suitable methods should be examined (especially in cluster analysis) and marker parameters.

Annotated reviews are not available for download in order to protect the identity of reviewers who chose to remain anonymous.

Reviewer 2 ·

Basic reporting

No comment

Experimental design

No comment

Validity of the findings

The RAPD method was used extensively in the 1990s due to its ease of application. It is no longer preferred due to its disadvantages such as such as the dominant markers and the difficulty of diagnosing heterozygotes, limited reliability, different results in different laboratories, different results even when switching from one thermocycler device to another, and difficulties in transferring the markers obtained in this way to other maps.
Therefore, I believe that removing the RAPD method from the article, including only the SSR results, and updating Figures 4 and 5 by determining the Optimal ΔK value will strengthen the scientific aspect of the article.

Additional comments

1. Line 62 the annual mango production unit is not specified.
2. Line 66 The statistics of the Ministry of Agriculture and Land Reclamation statistics of the Ministry of Agriculture belong to the year 2015. The current date of these statistics should be given.
3. Line 131 “thermos cycler” should be corrected as “thermo cycler”
4. Line 183, LMMA_4 is included in Figure 1b, but not in Table 2 and Table 5. Also, MIAC_5 is located in Figure 1b, not figure 1a. It should be corrected in the manuscript.
5. Line 226, starting with this sentence “The Shannon index indicates the degree of polymorphism among genotypes” should be a separate paragraph.
6. Line 358, the mean PIC value for RAPD is given as 0.89, but Table 4 shows that this value is 0.378.
7. The combined use of mango cultivars, or mango genotypes is caused confusion in the manuscript. Instead, I think it would be more correct to use of “cultivars” in whole manuscript.
8. Line 586, in Figure 1 legends spelling of “Seventeen” should be corrected as “seventeen”. There is also no need for Figure 1 legend on page 21.
9. In the dendogram graph in Figure 2, which shape is (a) and which shape is (b) is not specified. There is also no need for Figure 2 legend on page 23.
10. There is also no need for Figure 3 legend on page 25.
11. There is also no need for Figure 4 legend on page 27.
12. Line 596, in Figure 5 legends sentence of “Demographic structure of fig genotypes” should be corrected as “Demographic structure of mango genotypes”. There is also no need for Figure 5 legend on page 29.

Reviewer 3 ·

Basic reporting

1. For genetic analysis 17 Mango cultivars are inadequate.
2. in the main text ı mentioned in the abstract as please as much as you can the keywords should be in Title, this will help you to increase your citations indirectly). In the Abstract there is no genetic distance, genetic diversity, Pollyporhism information content (PIC), and so on).

Line 109-Line 111: What is the aim of this study? Why did you need to do such research, what kind of problem do you want to bring a solution? Is there a lack of genetic diversity? Does this restrict the breeding of the Mango plant?

The number of mango tree are adequate, for such an analysis, at least 60 genotypes should be used).

Experimental design

Please indicate the PCR cycles.

Amplification of SSR markers? Some details.

Pay attention to Line 169 highlighted words.

Line 187. How did you harvest, shouldn’t you use the wep-based harvest address?

Validity of the findings

There are 2 figures combined and named Figure 4. In the first one, the cultivars distribution is different from the second. Normally no matter which marker type you used, the reasons must be identical coz this distribution does not depend on the marker types, it depends on the ancestor. Please try to do reliable data scoring. I recommend the same thing for Figure 3, too.

Line 359-Line 360. Low phrase.
Line 360-362. Did you try to say that RAPD markers are more reliable than SSR? Is it possible that RAPD markers amplified more loci than SSR? Or Did you find the PanWAR et al., 2010 paper? It is really strong idea, and it need to be proven by the PANWAR et al., 2010).

Comparing the discussion section with relevant literature and highlighting the reasons behind similarities and differences can enhance the quality of the publication and shed light on science. The discussion section needs to be better structured to effectively convey the findings of the study and their implications.

Additional comments

You should have a solid suggestion in the conclusion. The fact that the Zebda variety is genetically different does not mean much. All genotypes are genetically different from each other. We identify the most distant relatives in genetic diversity analyzes and determine the varieties that will participate as parents in the variety development breeding programs, taking into account their resistance to yield quality, biotic and abiotic stress factors. In other words, since the varieties that are farthest from each other can produce more resistant offspring, the determination of the degree of kinship of the individuals constitutes the basis of the breeding program as it provides the opportunity to select the most ideal parent for the breeding program. Therefore, I believe that the conclusion part should be strengthened a little more in the light of this information.

Annotated reviews are not available for download in order to protect the identity of reviewers who chose to remain anonymous.

---

## Round 0.2 · Minor Revisions

Dear authors
Two reviewers expressed widely divergent views. However I disagree with Reviewer 2. The manuscript may be accepted by correcting his/her points.
I think the authors acted hastily and carelessly. Please correct the indicated places carefully. There are too many typos in your manuscript. Please resubmit your manuscript with more attentive and careful corrections.
Sincerely

Reviewer 1 ·

Basic reporting

Dear editor
The points raised in the first round of review have been largely followed by the authors and English has been improved to a great extent. And in my opinion, the manuscript is acceptable after minor revision.
some suggestion:
Comparative assessment remove from the list of keywords,
Abbreviations of the calculated indices and their formulas are given (Line 197-202).

Experimental design

The experimental design is appropriate.

Validity of the findings

It seems that the manuscript has the necessary standards after revision.

Reviewer 2 ·

Basic reporting

no comment

Experimental design

no comment

Validity of the findings

no comments

Additional comments

1. In line 84 “…are distributed 9nei, 1987). Is it reference?
2. In the first manuscript in material method section you said the leaf material was collected in 2019. But in the current manuscript you said that the leaf material was collected in 2021. Which one is true?
3. In the first manuscript I said to you the statistics of the Ministry of Agriculture and Land Reclamation should be updated. But you were changed only the year of the stats. Is there any change in statistics in 5 years?
4. Line 124 “Thirty of the microsatellite markers utilized in this investigation were previously described by…” the sentence is not finished.
5. Line 203 in the text you said that the PIC value of OPA_18 was 0.407, but in the Table 4, I see that it is 0.405, which one is true?
6. Line 208 in the text you said that the average of HAVP 0.43, but in the table 5 the average is 0.043, which one is true?

• There are too many typing errors in the manuscript,
• The authors ignoring previous revisions,
• The date of collection of plant samples in the previous manuscript and the dates in the current manuscript do not match and etc.
It shows that the authors do not take the revisions seriously and my confidence to the manuscript has been shaken. For these reasons, I refuse to publish the article in PeerJ, out of concern that it will reduce the quality of the journal.

---

## Round 0.3 · Minor Revisions

Dear authors,

There are still typographical errors in your manuscript. Please review carefully once again. I see two different fonts in the conclusion section. Correct and resubmit.

Best regards

---

## Round 0.4 · accepted · Accept

Your manuscript is now acceptable. Congratulations